# Spatiotemporal Variation and Prediction Analysis of Land Use/Land Cover and Ecosystem Service Changes in Gannan, China

Xin Luo, Yongzhong Luo *, Fangjun Le, Yishan Zhang, Han Zhang and Jiaqi Zhai

College of Forestry, Gansu Agricultural University, Lanzhou 730070, China; 19141995315@163.com (X.L.); 18394496784@163.com (F.L.); z865479918@163.com (Y.Z.); shiwochun540277@163.com (H.Z.); 18606468628@163.com (J.Z.)
* Correspondence: luoyzhong@gsau.edu.cn

**Abstract:** For the preservation of ecosystems, including the enhancement of ecological strategies, examining the temporal and geographical variance in ecosystem services (ESs) and land use/land cover change (LUCC) is crucial. Gannan is situated on the upper Yellow River, which is a notable water conservation region with excellent ecological quality, but in the background of the local traditional production mode and rapid economic development, natural disasters, grassland degradation, and other ecological problems occur frequently. The integrated valuing of ecosystem services and tradeoffs (InVEST) model and the patch-generating land use simulation (PLUS) model are combined in this work to assess the spatiotemporal variance in ESs in Gannan. We set up three scenarios in modeling future land use—ecological protection (EP), natural development (ND), and economic development (ED) in 2050—and analyzed and evaluated the drivers of the variation in ESs. In order to reveal the LUCC in Gannan between 1990 and 2020, we predicted the LUCC and ESs spatial distribution characteristics in 2050, explored the correlation between its driving factors, and comprehensively analyzed and propose optimization measures and protection strategies. Through several simulation experiments, the findings indicate the following: (1) the largest percentage of land expansion for construction in Gannan between 1990 and 2020 is 74.53%, and the most noticeable percentage of shrinkage in the sand area is 20.67%; (2) from 1990 to 2020, Gannan's water yield, carbon storage, soil retention, and habitat quality all changed, by $60 \times 10^8$ m$^3$, $0.04 \times 10^8$ t, $-10.66 \times 10^8$ t, and $-0.02$, respectively; (3) ESs are influenced by a variety of natural and societal variables: the southern and southwestern regions of Gannan are home to the majority of ESs hot spot areas, while the northern region is home to the majority of cold spot areas. This study contributes to the analysis of the developmental traits of Gannan ecosystems and can serve as a model for the preservation of terrestrial ecosystems with comparable environmental traits.

**Keywords:** land use/cover change; ecosystem services; spatial and temporal patterns; PLUS model; InVEST model

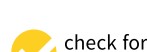


## 1. Introduction

Land use/land cover change (LUCC) not only influence the distribution and pattern of ecological risks, but also have important implications for socioeconomic development [1,2]. LUCC can intuitively reflect changes in landscape patterns within a region, especially in areas of high human activity, and can reflect forms of human–nature interaction. However, ecosystem services (ESs), which frequently include providing, supporting, regulating, and cultural services, are crucial to human survival and can seriously jeopardize human well-being if they are not given priority when decisions are made [3]. LUCC is a major factor in variations in the ESs' value, increasing their value and optimizing human benefits from them [4,5]. As a result, investigating the temporal and spatial dynamics of land use trends and logically and objectively assessing the ecosystem services level variations will

support the region's sustainable development by balancing the needs of the economy and environment [6].

The study of LUCC reflects changes in human utilization of Earth's resources and can also demonstrate spatiotemporal alterations in ecological environments at different scales; the relevant research has attracted widespread attention in different countries, ecological environments, and climatic zones [7]. In recent years, research combining LUCC with ESs has primarily focused on the assessment of ecosystem service value and function. For example, studies in Minnesota have compared the impacts of land use change on ecosystem services, biodiversity, and landowner returns. These studies have found that different land use choices can have varying effects on different ecosystem services [8]. Currently, research on this topic still faces many challenges. Ecological studies need to address a range of spatial and temporal scale issues, requiring the collection of diverse data sources and the utilization of various methods. Improvement in research is necessary from multiple perspectives including ecological, social, and economic aspects. This will ultimately facilitate the sustainable development of regional ecological environments and socioeconomics [9,10].

At this stage, there are a number of types of simulation software available to assist researchers in the simulation of LUCC and prediction. Different research directions require choosing more applicable models to assist in the research, such as the cellular automata (CA) model [11], CA–Markov model [12], CLUE-S model [13], future land use simulation (FLUS) model [5], and patch-generating land use simulation (PLUS) model [14]. In recent studies, the advantages of the PLUS model have become more and more obvious. In contrast to other simulation models, the PLUS model focuses more on identifying the mechanism underlying land use changes during the simulation process. This is achieved by integrating the land expansion analysis strategy with a CA model based on various types of stochastic patch seeds [15].

The concepts of ecosystem functions and ecosystem services are susceptible to confusion; the former emphasizes mechanisms sustaining ecological system integrity, while the latter underscores the expression of ecosystem contributions to human well-being [16]. Ecosystem service functions refer to the natural environmental conditions and benefits formed and sustained by ecosystem processes crucial for human survival [17]; therefore, selecting and quantifying ecosystem service functions are crucial steps in the relevant research. The integrated valuation of ecosystem services and tradeoffs (InVEST) model is the most widely used software at this stage. The InVEST model can well quantify the functioning of ecosystem services, and it is easy to operate and makes it relatively easy to collect the data. The InVEST model has played an important role in many research fields. Li et al. evaluated and examined the effects of socioeconomic development on ecosystem service functions, identified significant ecosystem service functions, and examined the connection between ecosystems and tourism development [18]. A foundation for evaluating the ecological advantages of converting farmland back to forest in the Yellow River Basin was established by Zhao et al.'s study, which employed the InVEST model to gauge the extent to which ecological protection measures affected the ecosystem [19]. The four essential ESs that follow were chosen, namely water yield, carbon storage, soil conservation, and habitat quality, and these were evaluated to be able to reflect the ecological indicators of climate change, vegetation change, land cover change, and multiple aspects of biodiversity.

The method of coupling the PLUS model with the InVEST model to simulate land use change and quantify key ecosystem services has become a widely used paradigm; valuable conclusions have been drawn in research areas of different ecosystems [20,21]. The main differences primarily manifest in the methods used to simulate and predict future land use processes. Currently, commonly used prediction methods include Markov chain analysis, the system dynamics (SD) model, multi-objective optimization (MOP), and gray multi-objective optimization (GMOP) [22–25]. The methods selected by different studies are based on the characteristics and actual conditions of the study area. By integrating the PLUS and InVEST models, simulating future land use data, and analyzing the resulting

changes, we can reveal the impact of land use on various ecosystem services and identify the drivers influencing their functions [26]. In modeling future land use, not only are historical climatic and socioeconomic data needed, but also relevant future data are needed as drivers to be input into the model. The sixth international Coupled Model Intercomparison Project (CMIP6) provides a variety of future global climate change scenarios to inform researchers [27]. Different from CMIP5, CMIP6 incorporates anthropogenic drivers of climate change and provides more comprehensive and accurate data for future scenarios [28]. Using the climate and socioeconomic data of the above development pathways as driving factors, combined with the policy context and ecological management approach, can provide a basis for predicting the outcomes of future land use simulations under various scenarios.

Ecological civilization construction is a model of ecological civilization development that China has been advocating for a long time, advocating that humans coexist peacefully with the environment [29]. China has implemented a number of initiatives to protect the ecological environment, which have achieved certain successes, and is moving forward with the concept of sustainable development [30]. Gansu Province, an important province in the inland northwest of China, has a long and narrow topography with complex and diverse landforms, and the evolution of the ecological environment is also more complex and diverse. Gannan is located in the southern part of Gansu, and its ecological environment is relatively good compared with other areas of Gansu, and the provisioning, supporting, and cultural services in the ecosystem services of Gannan Prefecture are at a high level, while the regulating services are only at a medium level [31]. Long-standing environmental and ecological issues resulting from the distinctive physical features and customary agricultural practices of the Gannan area have included soil erosion, deterioration of grasslands, and other issues [32]; therefore, there is still a need to find more effective initiatives for ecologically sustainable development in Gannan.

(1) We first analyzed the LUCC patterns and distribution in Gannan from 1990 to 2020. This foundational step provides the basis for subsequent simulations and analyses of future land use dynamics, along with the computation and assessment of ecosystem services. (2) The second step involved utilizing the PLUS model and its associated modules, integrating historical land use data. We conducted land use simulations for the year 2050 under various scenarios. (3) The subsequent step involved utilizing the InVEST model, integrating generated land use data. Adjustments to parameters related to water yield, carbon storage, soil retention, and habitat quality were made based on local conditions. This allowed for the analysis of the spatiotemporal distribution patterns of ESs in Gannan. (4) Finally, we simulated the spatiotemporal changes in the hot and cold spots of ESs; analyzing the hot and cold spots in Gannan can reveal spatial distribution patterns, indicating whether they exhibit clustering or dispersion and highlighting regions requiring attention or intervention [26,33]. Building upon the aforementioned points, we were able to analyze the spatiotemporal dynamics of land use and ecosystem services in Gannan. Subsequently, based on these findings, we propose recommendations for the ecological strategic development of Gannan.

## 2. Materials and Methods

### 2.1. Study Area

Gannan Tibetan Autonomous Prefecture is located in the southern region of Gansu Province, China ($100°46'$–$104°44'$ E, $33°06'$–$36°10'$ N), with a total area of about 38,521 km$^2$ (Figure 1). The region has a large difference in elevation, with the average elevation in the west, north, and south of the territory reaching more than 3000 m above sea level, and Guazigoukou in Zhouqu County being the lowest point. Gannan Prefecture is mainly characterized by three geomorphologic zones: the mountainous plains zone, canyon zone, and mountainous hills zone. As of 2022, the resident population is 683,700 and the GDP is 24.512 billion yuan.

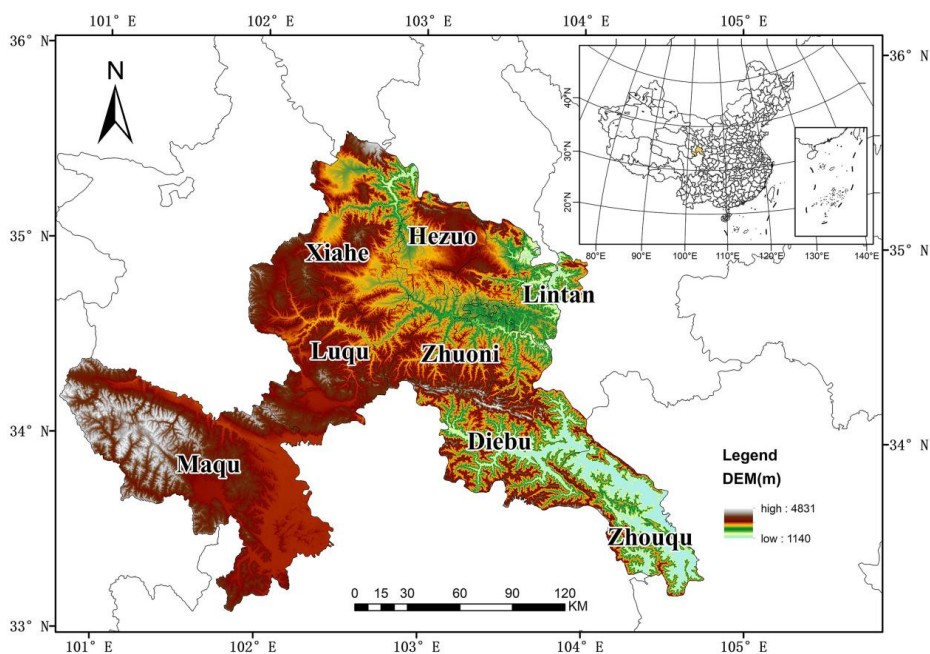

**Figure 1.** Gannan's geographical location.

Gannan has a high ecological status in China for its functions as a major national ecological area and an area that demonstrates advanced ecological civilization, an ecological security barrier of the country, and one of the first batch of areas demonstrating advanced ecological civilization in the whole country.

### 2.2. Data Collection and Processing

This paper collects and processes data from four dimensions: land use, physical geography, socioeconomics, and meteorology and climate data.

(1)  Land use data: Land use types for 1990 and 2020 are from the Resource and Environment Data Center of the Chinese Academy of Sciences (https://www.resdc.cn/, accessed on 16 June 2023), with a spatial resolution of 30 m.
(2)  Physical geographic data: The Geospatial Data Cloud (https://www.gscloud.cn/, accessed on 18 June 2023) provided the digital elevation model (DEM) data, on the basis of which slope degree and direction were analyzed and resampled to a resolution of 30 m; soil type data are from the China Soil Database (http://vdb3.soil.csdb.cn/, accessed on 16 June 2023), with a resolution of 30 m.
(3)  Socioeconomic data: Population and GDP data for historical scenarios are from the Resource and Environment Data Center of the Chinese Academy of Sciences (https://www.resdc.cn/, accessed on 30 June 2023); administrative boundary data, government sites, settlements, and roads are from the National Geographic Information Resource Catalog Service System (http://www.webmap.cn/, accessed on 16 June 2023); population and GDP data are from the Shared Socioeconomic Pathways (SSPs) Population and Economic Gridded Database [34].
(4)  Climatic data: Data on average annual temperature and rainfall were sourced from the National Tibetan Plateau Science Data Center (https://data.tpdc.ac.cn/, accessed on 20 June 2023) for both past and future scenarios [35].

The above impact factor data were collated and used as the base database for this study (Figure 2).

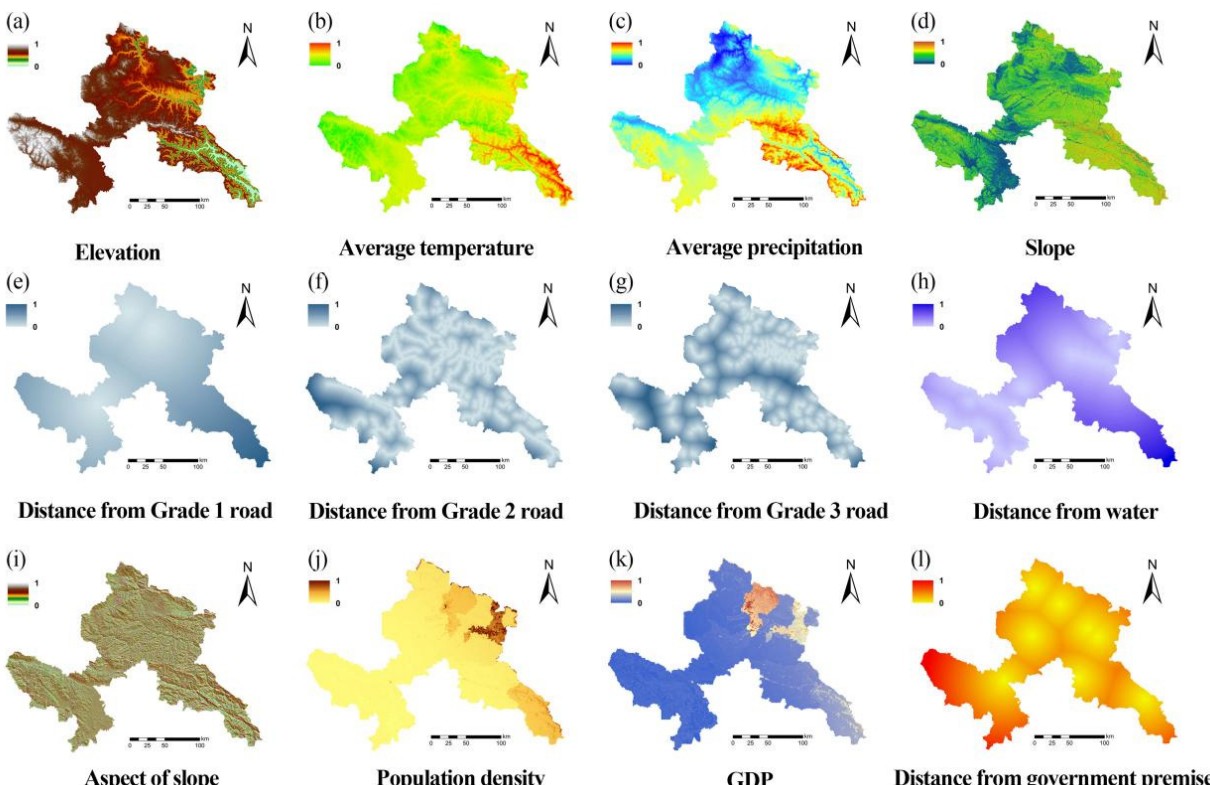

**Figure 2.** Main drivers of LUCC in Gannan. (**a**) Elevation. (**b**) Average temperature. (**c**) Average precipitation. (**d**) Slope. (**e**) Distance from Grade 1 road. (**f**) Distance from Grade 2 road. (**g**) Distance from Grade 3 road. (**h**) Distance from water. (**i**) Aspect of slope. (**j**) Population density. (**k**) GDP. (**l**) Distiance from grvernment premises. Note: Raw data for drivers were normalized to remove dimensional effects between factors.

### 2.3. Land Use Simulation

#### 2.3.1. The PLUS Model

This model improves the accuracy and efficiency of land use simulation by combining the land expansion analysis strategy (LEAS) rule mining framework with the multi-class random patches with type random seeds (CARS) CA model. Multiple future scenario data can be input as drivers for future planning [36]. The locations that changed were identified by the LEAS model. Using the random forest method, a sample was selected from these locations in order to investigate the association between each site type and driver. The conversion probability for each site was then determined using the following formula:

$$P_{i,k(X)}^d = \frac{\sum_{n=1}^M I(h_n(X) = d)}{M} \tag{1}$$

where $P_{i,k(X)}^d$ is the extended probability of land type $k$ at pixel $i$, $d$ indicates the existence of various types of land $k$ transfers to the land type, taking a value of 0 or 1, $X$ is a vector made up of driving factors, $h_n(X)$ is the terrain type determined by decision tree $n$, $M$ represents the total number of decision trees, and $I$ is the indicator function of the decision tree.

To simulate the spatial evolution of each classification, we integrated the traditional cellular automaton (CA) model with a patch generation and threshold reduction mechanism, creating the CARS model. This model facilitates the generation of land use development potential for each classification. When the neighborhood effect of a single category reaches

zero, the PLUS model autonomously generates a "seed" for each category, initiating the formation of a new patch using the following formula:

$$\Omega_{i,k}^{t} = \frac{con\left(c_i^{t-1} = k\right)}{n \times n - 1} \times w_k \tag{2}$$

$$OP_{i,k}^{1,t} = \begin{cases} P_{i,k}^1 \times (r \times \mu_k) \times D_k^t & \Omega_{i,k}^t = 0 \ and \ r < P_{i,k}^1 \\ P_{i,k}^1 \times \Omega_{i,k}^t \times D_k^t & Others \end{cases} \tag{3}$$

where $\Omega_{i,k}^t$ is the domain weight of ground class $k$ in pixel $i$ at moment $t$, $w_k$ is the domain weight parameter, $OP_{i,k}^{1,t}$ is the integration probability of the transition from pixel $i$ to ground class $k$ at moment $t$, $P_{i,k}^1$ is the fitness probability of the extension of pixel $i$ to ground class $k$, $D_k^t$ is the adaptive drive coefficient, $r$ is a random value between 0 and 1, and $\mu_k$ is the threshold for newly generated plaques.

### 2.3.2. Kappa Coefficient

The *Kappa* coefficient was used to validate the PLUS model's performance and confirm the land use simulation's correctness. The following formula yields the *Kappa* coefficient:

$$Kappa = \frac{P_o - P_c}{P_d - P_c} \tag{4}$$

where $P_o$ is the proportion of correct simulations, and $P_c$ is the expected scale of the simulation projected. $P_d$ is the ideal analog value, generally defined as 1; when *Kappa* > 0.8, it demonstrates that the model's correctness is statistically acceptable and the simulation results are reliable [15].

### 2.3.3. Scenario Design

Based on the new combination of the shared socioeconomic pathway and representative concentration pathway (SSP-RCP), three sets of future climate and socioeconomic data, SSP119, SSP245, and SSP585, were used to set up three different scenarios.

(1) Before the simulation, an accuracy test was conducted. A kappa value of more than 0.8 suggested that the model was appropriate for simulating land use. The number of pixels of used land in 2050 was predicted using Markov chain analysis; the generated number of pixels could provide a fundamental reference during the predictive simulation phase.

(2) Within the PLUS model's LEAS module, the constraints were inputted, while the yearly averages of rainfall, temperature, and population, and the GDP data of the three future scenarios (SSP119, SSP245, SSP585) were used as the future planning data. The contribution degree of the land use driving factors and the probability distribution of land development were derived.

(3) The CARS module was executed in the PLUS model, and in the transfer matrix settings, the value of 1 indicates that the transfer is permitted, while 0 signifies that the transfer is restricted, and the domain factors were set with reference to the relevant literature and the actual situation. In the process of forecasting the future land use demand in 2050, the number of pixels predicted by the Markov chain method in the PLUS model was used as the parameter input for the other scenarios of LUCC. The three scenarios are the SSP-119 ecological protection scenario (EP), the SSP-245 natural development scenario (ND), and the SSP-585 economic development scenario (ED) [37].

*2.4. InVEST Model*

2.4.1. Water Yield (WY) Module

The amount of water in an ecosystem is related to human life and influences the transformation of human life and ecosystems in many ways. The formula is as follows:

$$Y_{xj} = \left(1 - \frac{AET_{xj}}{P_x}\right) P_x \tag{5}$$

where $Y_{xj}$ is the water yield per year (mm) of land class $j$ in pixel $x$, $P_x$ is the average annual precipitation (mm) in pixel $P_x$, and $AET_{xj}$ is the actual evapotranspiration (mm) of land class $j$ in pixel $x$.

2.4.2. Carbon Storage and Sequestration (CSS) Module

The carbon that ecosystems store in their plants and soils is known as carbon storage services. The four types of carbon stocks that make up carbon storage services are soil, dead organic carbon, above-ground carbon, and below-ground carbon, according to the InVEST carbon module. After the identification of carbon pools in the study area with reference to multiple works in the literature [38–41], carbon pools were calculated in conjunction with land use data using the following formula:

$$C_{\text{total}} = \sum_{k=1}^{n} A_k \times (C_{\text{above}} + C_{\text{below}} + C_{\text{soil}} + C_{\text{dead}}) \tag{6}$$

where $C_{\text{total}}$ is the total carbon stock (t), $A_k$ is the area of land type $k$, $k$ is 1 to $n$, $n$ is the quantity of land types, and above-ground vegetation carbon density ($C_{\text{above}}$), below-ground vegetation carbon density ($C_{\text{below}}$), soil carbon density ($C_{\text{soil}}$), and dead organic matter carbon density ($C_{\text{dead}}$) constitute the carbon pools.

2.4.3. Sediment Delivery Ratio (SDR) Module

The ability of a region to stop soil erosion is measured by the sediment delivery ratio, and the calculations take into account the influence of sand trapping forces in the sample plots:

$$SR = R \times K \times LS - R \times K \times LS \times C \times P \tag{7}$$

where $SR$ is the total amount of soil retained throughout the year (t·hm$^{-2}$·a$^{-1}$), $LS$ is the terrain factor calculated from the slope length factor ($L$) and slope factor ($S$), $K$ is the soil erodibility factor, $C$ is the vegetation cover factor, $P$ is a factor of soil conservation measures, and $R$ is the rainfall erosivity index based on monthly precipitation.

2.4.4. Habitat Quality (HQ) Module

The habitat quality module of the InVEST model was used to calculate the habitat quality index (HQI). Habitat quality reflects the degree of suitability of the ecological environment for human survival and the sustainable development of society and the economy, which can be written as the following equation:

$$Q_{xj} = H_j \left(1 - \left(\frac{D_{xj}^z}{D_{xj}^z + K^z}\right)\right) \tag{8}$$

where $Q_{xj}$ is the habitat quality of land type $j$ in pixel $x$, taking values from 0 to 1. $H_j$ is the suitability of land type $j$, $K^z$ is the model parameter as a constant, and $D_{xj}^z$ is the degradation of land type $j$ in pixel $x$.

*2.5. Technical Lines of Research*

In this paper, research related to LUCC and ESs function in Gannan was carried out through five steps: (1) The first step involved collecting relevant natural and social data and preprocessing the data to standardize the accuracy and create a database for geographic

analysis of the study area. (2) The land use geographical distribution was predicted using the PLUS model under several scenarios in 2050 by combining the historical LUCC through the Markov chain method and inputting the relevant scenario parameters. (3) Historical LUCC was used to simulate future LUCC data to be input into the InVEST model as basic parameters, key ESs were selected, and specific parameters of each ES were adjusted by combining the results of previous research to generate the ESs in Gannan. (4) Finally, we computed the comprehensive scores of ESs to generate the changes in the spatial and temporal distribution of ESs in Gannan, analyzed the driving factors of LUCC and ESs and investigated their correlations, established the weights of ESs using the entropy weighting method, and provided recommendations for the sustainable development strategy of Gannan based on the research findings (Figure 3).

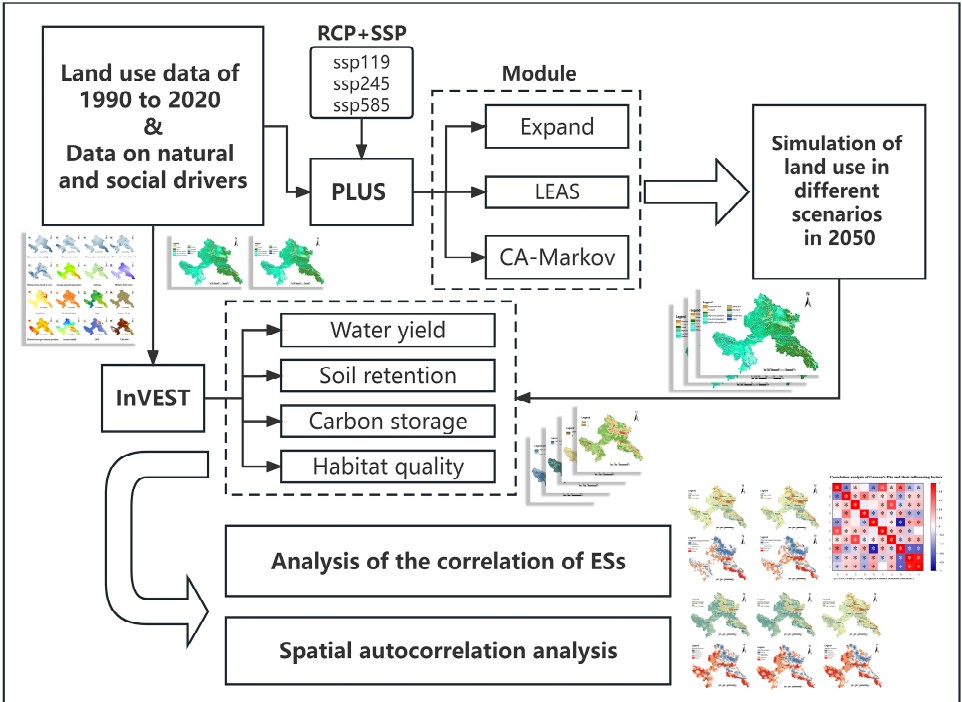

**Figure 3.** Research methodological framework.

## 3. Results

### 3.1. Spatial and Temporal Distribution Pattern of Land Use in Gannan

The spatiotemporal distribution of LUCC between 1990 and 2020 as well as the land use patterns in space and time in scenarios of different periods (2020–2050) were analyzed and simulated, respectively. Before simulating the distribution of land use in future scenarios, the PLUS model needs to be tested for accuracy. Using 16 driving factors as constraints, starting from 1990, the simulation yielded 2020 land use data, and then the 2020 land use simulation data were tested against the real 2020 land use data through the validation module, and the running results of the kappa coefficient reached 0.905, with an overall accuracy of 0.923.

### 3.1.1. LUCC in Gannan under Historical Scenarios

Considering Gannan Prefecture's historical land use distribution pattern (1990–2020), the transfer of land use types mainly occurred in the southwestern and northern regions, with obvious spatial heterogeneity (Figure 4a); until 2020, the grassland land use type in Gannan Prefecture occupied an area of 20,808.07 km$^2$ accounting for 56.75%, which makes it the land use type that occupies the largest area; then, with an area of 11,257.73 km$^2$, forest land accounted for 30.71% of the entire land use types, and the smallest was the sandy land area, which accounted for 0.03%, and 70.59 km$^2$.

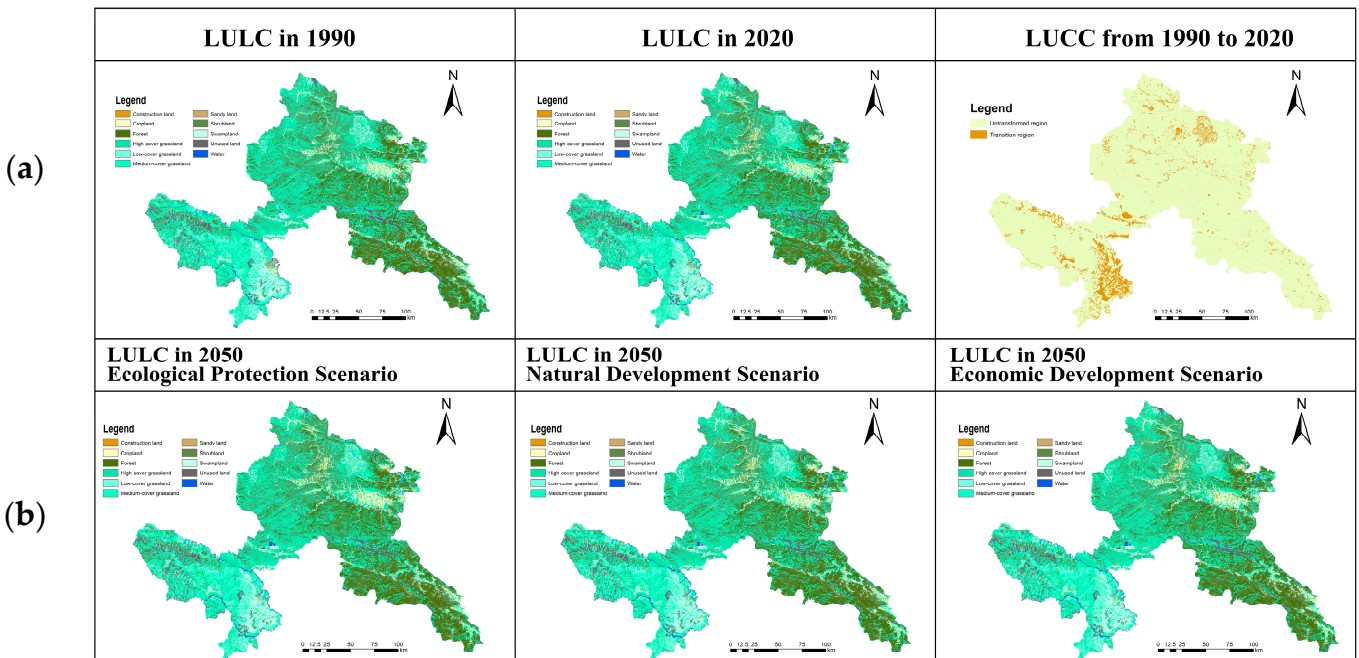

**Figure 4.** Land use change in 1990–2020 and land use modeling for 2050. (**a**) LUCC from 1990 to 2020. (**b**) LUCC in 2050.

Medium-cover grasslands, high-cover grasslands, shrublands, swamps, and sandy areas decreased more, by 207.08 km$^2$, 48.73 km$^2$, 89.2 km$^2$, 49.3 km$^2$, and 31.66 km$^2$, respectively. Cropland, low-cover grassland, watersheds, and construction land expanded by a large area, increasing by 117.23 km$^2$, 237.25 km$^2$, 39.65 km$^2$, and 73.2 km$^2$, respectively. The areas of forested and unutilized land did not change much relative to the overall land use transfer area, increasing by 10.6 km$^2$ and 8.04 km$^2$, respectively.

With regard to the proportion of area transferred by LUCC, the most significant proportion of land used for construction was expanded, with an increase of 74.53%, while the percentages of cropland, low-cover grassland, and water increased by 7.99%, 17.94%, and 17.52%, respectively, and the proportion of sandy land area was reduced most significantly, with a decrease of 20.67%. Other land use types were also transferred to different degrees (Table 1).

**Table 1.** LULC in the study area from 1990 to 2020 (area unit: km$^2$).

| Land Use/Cover Type | 1990 | 2020 | Change (1990–2020) | |
|---|---|---|---|---|
| | | | Transfer Area | Transfer Proportion |
| Cropland | 1467.67 | 1584.9 | 117.23 | 7.99% |
| Forest | 5864.48 | 5875.08 | 10.6 | 0.18% |
| Shrubland | 5471.85 | 5382.65 | −89.2 | −1.63% |
| High-cover grassland | 11,231.54 | 11,182.81 | −48.73 | −0.43% |
| Medium-cover grassland | 8332.54 | 8065.46 | −267.08 | −3.21% |
| Low-cover grassland | 1322.55 | 1559.8 | 237.25 | 17.94% |
| Water | 221.27 | 260.92 | 39.65 | 17.92% |
| Construction land | 98.21 | 171.41 | 73.2 | 74.53% |
| Swampland | 1426.55 | 1377.25 | −49.3 | −3.46% |
| Sandy land | 102.25 | 70.59 | −31.66 | −30.96% |
| Unused land | 1124.79 | 1132.83 | 8.04 | 0.71% |
| Total | 36,663.71 | 36,663.71 | | |

The main distribution of land uses is as follows: agriculture is primarily found in areas with minor topographic height differences in Zhoni and Lintan Counties; forests

are primarily found in Diebu and Zhouqu Counties; sandy land is primarily found in the eastern portion of Maqu County; and construction land is primarily found in the city of Hezuo.

### 3.1.2. LUCC in Gannan under Future Scenarios

Different scenarios were selected to combine different expansion probabilities according to CMIP6. Future scenarios for GDP, temperature, precipitation, and population growth as well as SSP119, SSP245, and SSP585 were incorporated for the 2020–2050 timeframe (Figure 4b). Representing the EP scenario, the ND scenario, and the ED scenario, respectively, the pattern of the number of pixels and land use types in the simulation results changed to a certain extent, and the scenarios were driven by future meteorological and social data, and there were some differences between the scenarios (Table 2).

**Table 2.** LULC in the study area from 2020 to 2050 (area unit: km$^2$).

| Land Use/Cover Type | 2020 | 2050 | | | Change (2020–2050) | | | | | |
|---|---|---|---|---|---|---|---|---|---|---|
| | | ND | EP | ED | ND | EP | ED | ND | EP | ED |
| Cropland | 1584.9 | 1689.36 | 1589.68 | 1689.36 | 104.46 | 4.78 | 104.46 | 6.18% | 0.30% | 6.18% |
| Forest | 5875.08 | 5884.36 | 5884.36 | 5781.62 | 9.28 | 9.28 | −93.46 | 0.16% | 0.16% | −1.62% |
| Shrubland | 5382.65 | 5340.58 | 5653.06 | 5296.45 | −42.07 | 270.41 | −86.2 | −0.79% | 4.78% | −1.63% |
| High-cover grassland | 11,182.81 | 11,140.62 | 11,140.62 | 11,140.62 | −42.19 | −42.19 | −42.19 | −0.38% | −0.38% | −0.38% |
| Medium-cover grassland | 8065.46 | 7812.53 | 7812.53 | 7915.27 | −252.93 | −252.93 | −150.19 | −3.24% | −3.24% | −1.90% |
| Low-cover grassland | 1559.8 | 1779.5 | 1530.54 | 1779.5 | 219.7 | −29.26 | 219.7 | 12.35% | −1.91% | 12.35% |
| Water | 260.92 | 298.74 | 298.74 | 298.74 | 37.81 | 37.81 | 37.81 | 12.66% | 12.66% | 12.66% |
| Construction land | 171.41 | 172.57 | 174.93 | 240.48 | 1.15 | 3.52 | 69.07 | 0.67% | 2.01% | 28.72% |
| Swampland | 1377.25 | 1346.16 | 1409.86 | 1330.26 | −31.09 | 32.61 | −46.99 | −2.31% | 2.31% | −3.53% |
| Sandy land | 70.59 | 60.08 | 52.19 | 52.19 | −10.51 | −18.39 | −18.39 | −17.49% | −35.24% | −35.24% |
| Unused land | 1132.83 | 1139.21 | 1117.2 | 1139.21 | 6.38 | −15.63 | 6.38 | 0.56% | −1.40% | 0.56% |
| Total | 36,663.71 | 36,663.71 | 36,663.71 | 36,663.71 | | | | | | |

Simulated with PLUS software, compared with 2020, the area of cropland in Gannan has the largest expansion area under the ND and ED scenarios, with an expansion area of 104.46 km$^2$, an increase of 6.18%; forested land decreases by 179.66 km$^2$ under the ED scenario, a decrease of 1.63%; shrubland grows very significantly under the EP scenario, with an expansion area of 270.41 km$^2$, which increased by 4.78%; construction land does not change much under the ND and EP scenarios, and expands very significantly under the ED scenario, with an increased area of 69.07 km$^2$, which is an increase by 28.72% of the original area. In addition to this, the grassland area is larger in the ND and ED scenarios than in the EP scenario; sandy land shrinks notably in each of the three situations; and three land use types, watershed, swamp, and unutilized land, do not change much in the three simulation scenarios (Figure 5).

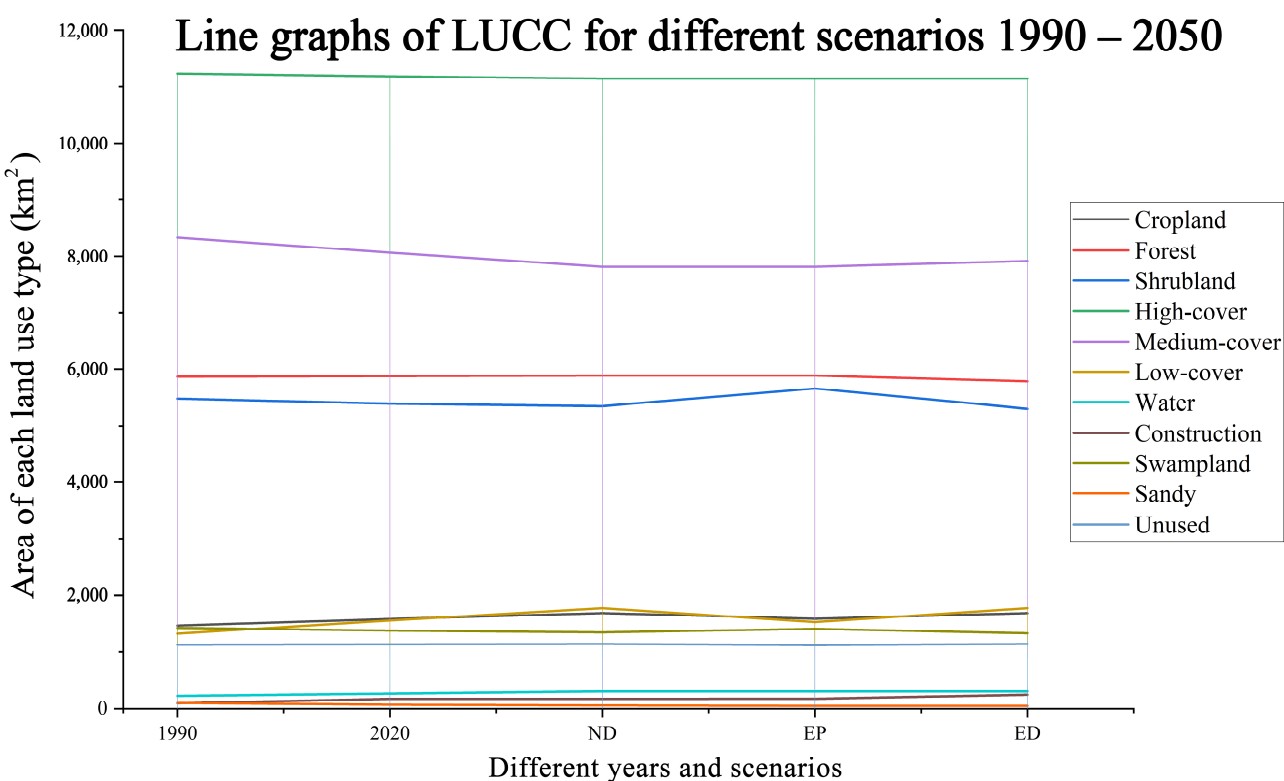

**Figure 5.** The changing trends in the areas of various land use types.

*3.2. Spatial and Temporal Transfer of Land Use in Gannan from 1990 to 2050*

In the research area's LUCC between 1990 and 2020, the LUCC occurred mainly between grassland, forest, and cropland. Cropland was mainly transferred to grassland and construction land, with the areas transferred being 97.55 km$^2$ and 38.76 km$^2$. Forested land and shrubland transferred the most area to high-cover grassland, 63.71 km$^2$ and 126.36 km$^2$, respectively. Swampland transferred 39.51 km$^2$ to high-cover grassland and 15.01 km$^2$ to water; sandy land was mainly transferred to medium-cover grassland and low-cover grassland, with areas of 21.33 km$^2$ and 15.18 km$^2$, respectively. Overall, during the period 1990–2020, the transfer of land use among grasslands with different coverage is more balanced, while the transfer of grasslands to forests and shrublands is larger, while the transfer of sandy land is mainly to grasslands, which to a certain extent can reflect that Gannan's ecological protection has been quite effective during the past three decades.

The conversion between land uses under the future scenarios mainly occurs between forest land, grassland, swampland, and cropland, and the area undergoing conversion (both outgoing and incoming) exceeds 5% of all land transfer areas, except for cropland under the EP scenario. Under the EP and ND scenarios, the expansion of shrubland mainly originates from the conversion of medium- and high-coverage grassland, with conversion areas of 265.93 km$^2$ and 62.21 km$^2$, respectively; the construction land's expansion occurs in the economic development scenario, and mainly originates from medium- and high-coverage grassland, with conversion areas of 102.74 km$^2$ and 76.85 km$^2$, respectively; the construction land's expansion occurs in the ED scenario, mainly from arable land and grassland, and medium-coverage grassland has the largest area transferred, namely 50.17 km$^2$ (Figure 6).

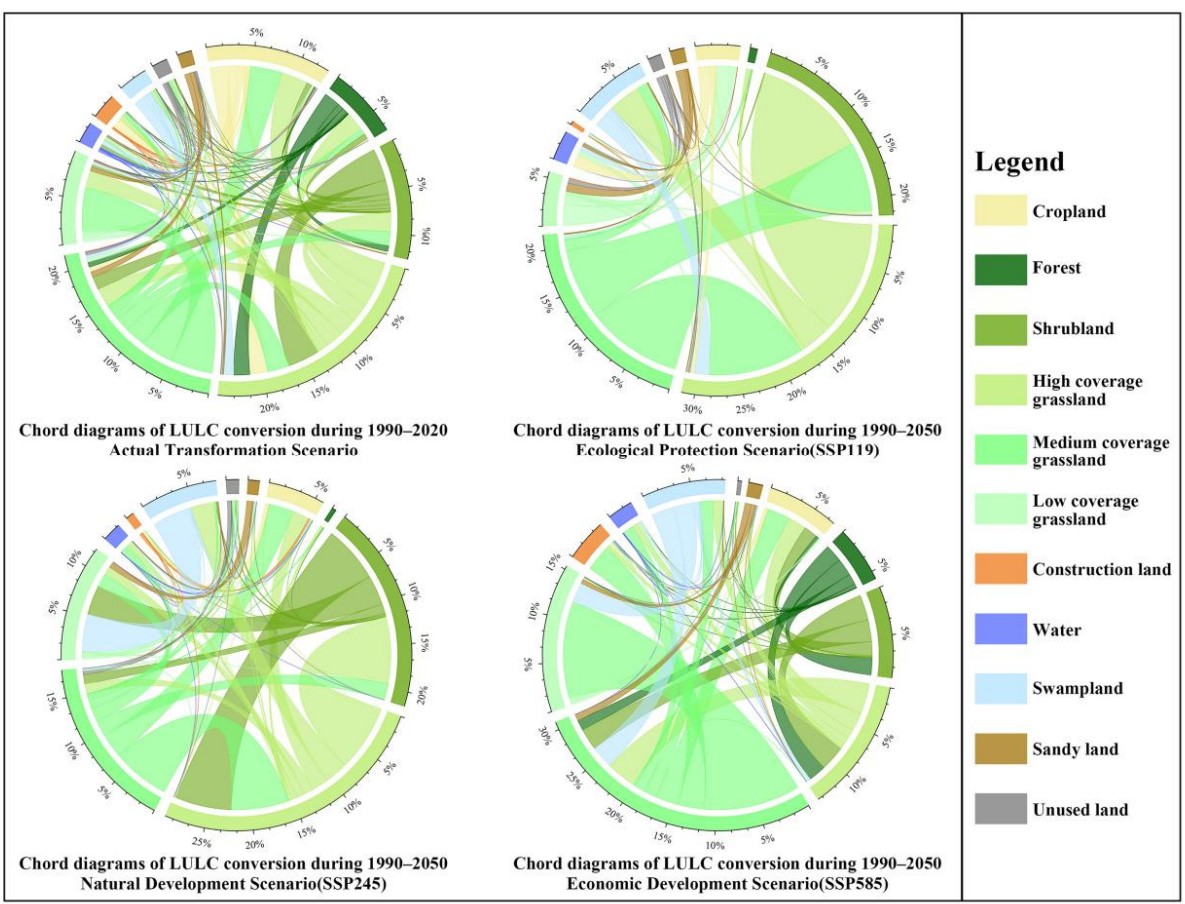

**Figure 6.** Gannan 1990 to 2050 land use transfer chord map.

### 3.3. Spatial and Temporal Changes in ESs in Gannan

3.3.1. Spatial Distribution Pattern of ESs in Gannan

Water yield and habitat quality revealed a declining tendency in the geographical distribution from west to east, and the regional distribution of soil retention and carbon storage revealed a declining tendency from east to west. Particularly affected by altitude and topography, the spatial heterogeneity of southwestern and southeastern Gannan from central and northern Gannan is obvious. The ecological environment of Gannan is generally the best in Gansu Province; however, modern agriculture has not advanced to a great extent in Gannan, and the natural environment is vital to many sectors, particularly agriculture. The degree of ecosystem services is crucial for local development given its physical limitations and reliance on conventional production methods [42].

The main factors influencing Gannan's water yield are evapotranspiration and annual precipitation. High-value regions of water yield are primarily distributed in the western part of Maqu County and the junction of Diebu and Zhouqu Counties; in the eastern part of Maqu County, there are relatively significant differences in spatial distribution patterns (Figure 7a). The water yield was $44.23 \times 10^8$ m$^3$ in 1990 and $104.23 \times 10^8$ m$^3$ in 2020. Under different scenarios in 2050, the EP scenario yields a relatively high water yield of $55.15 \times 10^8$ m$^3$, while the ND scenario and the ED scenario do not have a big difference in yield, with values of $38.26 \times 10^8$ m$^3$ and $38.99 \times 10^8$ m$^3$, respectively.

High-value regions of carbon stocks are found in woodland and shrubland, while the low-value regions are primarily found in cropland and unused land. The overall pattern of carbon stock distribution is quite similar to that of land use distribution, with total carbon stocks declining from east to west (Figure 7b). The values were $7.09 \times 10^8$ t and $7.05 \times 10^8$ t in 1990 and 2020, and the simulated carbon stocks are $7.06 \times 10^8$ t in 2050 for the EP scenario, $7.01 \times 10^8$ t for the ND scenario, and $7.00 \times 10^8$ t for the ED scenario.

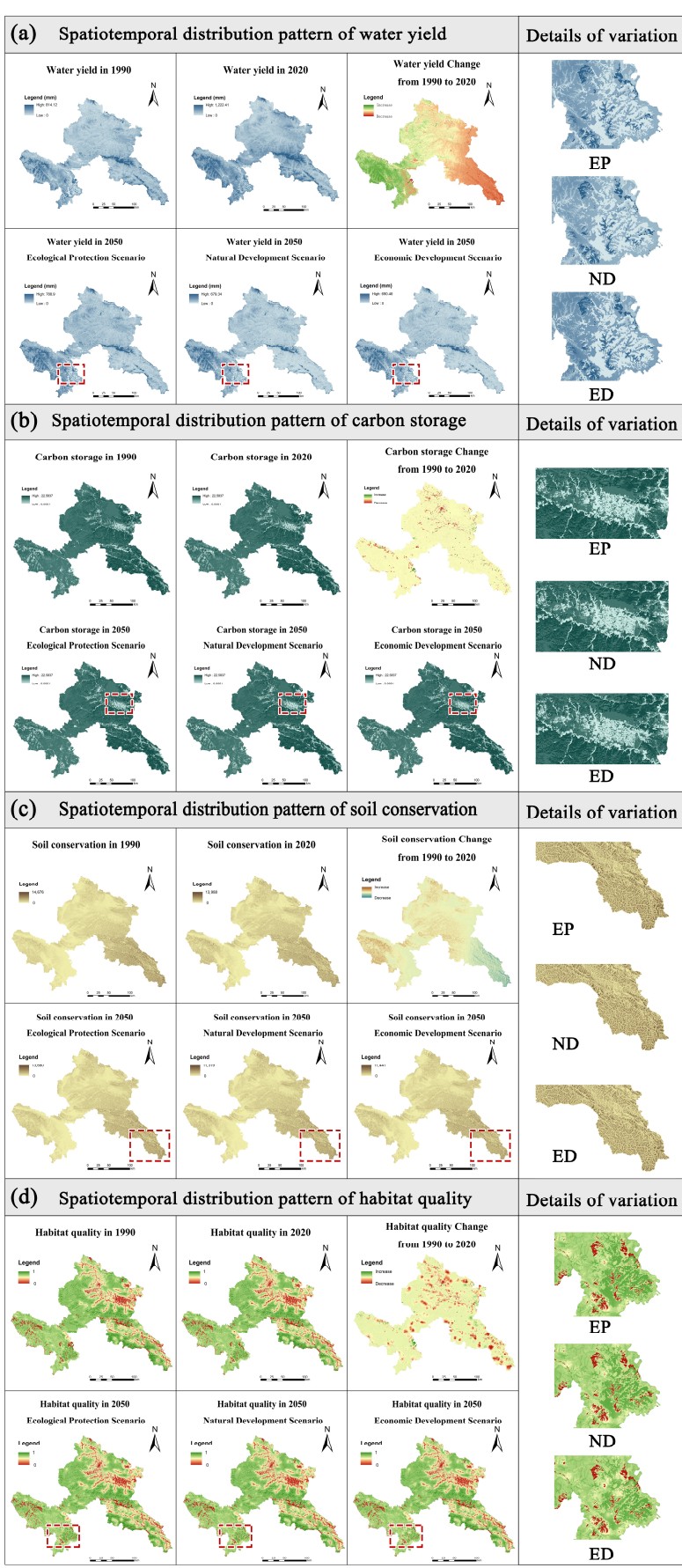

**Figure 7.** Spatial and temporal variation in ESs in Gannan. (**a**) Water yield. (**b**) Carbon storage. (**c**) Soil retention. (**d**) Habitat quality.

The spatial distribution pattern and change trend of soil retention are mainly closely related to elevation, topography, and rainfall, having a westward-high and eastward-low trend, with relatively high soil retention in Zhouqu and Diebu Counties, but with a decreasing trend in soil retention during the period from 1990 to 2020 (Figure 7c). Soil retention was $72.46 \times 10^8$ t and $83.12 \times 10^8$ t in 1990 and 2020, respectively, and is simulated to be $79.40 \times 10^8$ t in the EP scenario, $69.72 \times 10^8$ t in the ND scenario, and $67.66 \times 10^8$ t in the ED scenario in 2050.

Habitat quality is significantly affected by human activities, with the northern part of Gannan, where the city's economic development is more prosperous, having relatively low habitat quality due to intense human activities, and the woodland and grassland areas with less human intervention, especially in Maqu County, where rainfall is concentrated and sunshine is abundant, having relatively high habitat quality (Figure 7d). Average habitat quality was 0.78 and 0.76 in 1990 and 2020, respectively, and was modeled to be 0.77 for the EP scenario, 0.76 for the ND scenario, and 0.76 for the ED scenario in 2050.

### 3.3.2. Spatial and Temporal Trends in ESs in Gannan

Between 1990 and 2020, the water production increased by $60 \times 10^8$ m$^3$, and was found to decrease sharply again by 2050 through PLUS model and InVEST model simulation, with a relatively high water production of $55.15 \times 10^8$ m$^3$ under the EP scenario and relatively low water production of $38.26 \times 10^8$ m$^3$ and $38.99 \times 10^8$ m$^3$ under the ND and ED scenarios, respectively; the carbon stock changed with little magnitude, with the highest being $7.09 \times 10^8$ t in 1990 and the lowest being $7.00 \times 10^8$ t in 2050 under the ED scenario; the carbon stock changed with little magnitude, $10^8$ m$^3$ and $38.99 \times 10^8$ m$^3$, respectively; carbon stocks changed little, with the highest value being $7.09 \times 10^8$ t in 1990 and the lowest value being $7.00 \times 10^8$ t in 2050 under the ED scenario; soil retention increased significantly by $10.66 \times 10^8$ t under the historical scenario, and differed markedly between the EP and the remaining two scenarios under the future scenario; and habitat quality varied little between the historical and simulation periods in total, ranging from 0.76 to 0.78 (Table 3).

**Table 3.** Average values of ecosystem services from 1990 to 2050.

| Type | 1990 | 2020 | 2050 (EP) | 2050 (ND) | 2050 (ED) |
|---|---|---|---|---|---|
| Water yield (m$^3$) | $44.23 \times 10^8$ | $104.23 \times 10^8$ | $55.15 \times 10^8$ | $38.26 \times 10^8$ | $38.99 \times 10^8$ |
| Carbon storage (t) | $7.09 \times 10^8$ | $7.05 \times 10^8$ | $7.06 \times 10^8$ | $7.01 \times 10^8$ | $7.00 \times 10^8$ |
| Soil retention (t) | $72.46 \times 10^8$ | $83.12 \times 10^8$ | $79.40 \times 10^8$ | $69.72 \times 10^8$ | $67.66 \times 10^8$ |
| Habitat quality | 0.78 | 0.76 | 0.77 | 0.76 | 0.76 |

The results show that habitat quality and carbon storage are higher in 1990 compared with 2020, and water yield and soil retention are lower compared with 2020; among the ESs in 2020, water yield is significantly higher than that in 1990 and the simulation prediction years. The simulation results largely align with the actual data presented in the Gannan Statistical Yearbook. Compared with 1990, habitat quality and carbon stock decreased, while soil retention significantly increased. In the future scenario, the ESs in the EP scenario exhibited notably higher values than those in the ND and ED scenarios. Although the differences in ESs values between the ND and ED scenarios were small, the overall ecological condition appeared degraded compared with that in the EP scenario.

### 3.4. Factors Influencing Changes in Ecosystem Services

Combined with the above results, the correlation between the ESs and the factors needs to be further analyzed. In the study area, 2000 points were randomly generated as sampling samples, these points were assigned values, a total of 1956 points were calculated after removing outliers, Spearman's correlation analysis was used, and the results showed that most of the correlations among ESs, between ecosystems and influencing factors, and

among influencing factors were correlated and that correlations among ecosystem service functions, between ecosystems and influencing factors, and among influencing factors were correlated.

(1) The correlation between annual average precipitation and water yield is the strongest, and elevation factors also have an impact. These factors are all positively correlated with water yield. In contrast, annual average temperature, population density, and GDP are negatively correlated with water yield, to some extent reflecting that human activities are not conducive to an increase in water yield. (2) Carbon storage is simultaneously influenced by natural and social factors. In comparison with other influencing factors, it shows a positive correlation with all factors except for elevation, which exhibits a negative correlation. It can be observed that the distribution pattern of carbon storage is highly similar to the distribution pattern of DEM data. (3) Soil retention is primarily influenced by slope, and concurrently, in low-altitude regions, the soil retention is higher. The research findings indicate a positive correlation between the soil retention and average precipitation, population density, and GDP. However, the correlation with annual average temperature is not statistically significant. (4) Habitat quality demonstrates a positive correlation with annual precipitation and elevation, while manifesting negative correlations with other influencing factors. The quality of habitats is intricately linked to human activities, with regions characterized by higher population density and elevated GDP levels leading to a reduction in the habitat quality index (HQI). (5) There are also some correlations among various influencing factors, with a notably significant negative correlation observed between annual average temperature and elevation. Conversely, a notably significant positive correlation was observed between population density and GDP (Figure 8).

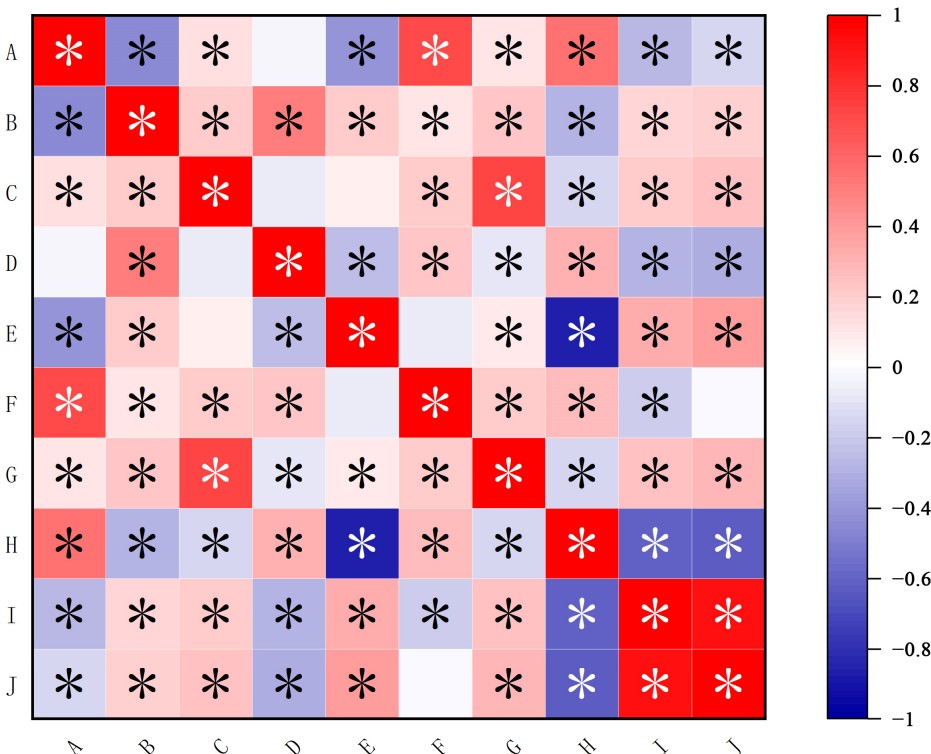

**Correlation analysis of Gannan's ESs and their influencing factors**

\* $p \leq 0.001$ (When $p \leq 0.001$, it signifies a notably substantial correlation.)

**Figure 8.** Correlation plot between ESs and influencing factors. (A) Water yield. (B) Carbon storage. (C) Soil retention. (D) Habitat quality. (E) Average temperature. (F) Average precipitation. (G) Slope. (H) Elevation. (I) Population density. (J) GDP.

Furthermore, connections were found amongst ESs, with water yield significantly harmonizing with soil retention and trading off with carbon storage; carbon storage having a strong synergy with soil retention and habitat quality; and habitat quality having a significant synergy with carbon storage and a nonsignificant correlation with the other two ESs [43].

## 4. Discussion

### 4.1. Spatial and Temporal Variation in Hot and Cold Spots in Gannan ESs

In order to carry out an exhaustive analysis of the effects of Gannan ESs on the ecological environment, four indicators were assigned weights using the entropy weighting method: water yield (0.33), carbon storage (0.15), soil retention (0.17), and habitat quality (0.35). This resulted in the evaluation results of Gannan ESs, and an additional analysis was conducted to examine the changes in the temporal and spatial distribution of cold and hot spots in Gannan ESs.

Overall, in 1990 and 2020, the overall mean values of ESs were 0.4134 and 0.4362, and the Moran I indices of 0.3858 and 0.4189 were both greater than 0, revealing that Gannan's ESs distribution was concentrated. Further Getis-Ord Gi* analysis was conducted and based on the Z score, the results showed that the cold spot areas were mainly concentrated in the northern region of Gannan, especially in the areas of Hezuo City and Lintan County, which have frequent human activities, faster economic growth, and greater intervention in the environment, so that the major cold spots of ESs were clustered in this region. Hot spots and sub-hot spots are mainly distributed in southwestern and southeastern Gannan, especially in the southern parts of Zhouqu County and Zhuouni County; the distribution of ecosystem hot and cold spots was more scattered in 1990 than in 2020, and the clustering of ESs was more pronounced in 2020 (Figure 9a).

In the future 2050 EP, ND, and ED scenarios, the overall mean values of ESs are 0.4220, 0.4080, and 0.4063, respectively, the level of ESs is optimal in the EP scenario, and the Moran I indices of 0.4022, 0.4134, and 0.4133, are all greater than 0, reflecting a significant clustering distribution of ESs under the simulation scenarios. The distribution of ESs in the simulation scenario is significantly clustered. The distribution of hot spots moved eastward compared with 2020, with more hot spot and sub-hot spot areas and fewer cold spot areas, while the distribution of hot and cold spots was also more concentrated. In the EP scenario, cold and hot spot distribution is more decentralized overall, but the mean value of ESs is higher (Figure 9b).

Overall, we should pay close attention to the ecological conditions of Hezuo City and Lintan County, reduce human overexploitation of natural resources, and also address the ecological risks present in Diebu County and Zhouqu County. In other counties, it is necessary to continue adhering to relevant ecological conservation policies, intensify protection efforts in hot spot areas of ecosystem services, extend environmental improvement to surrounding areas, and further enhance the overall ecological status of Gannan.

### 4.2. LUCC and ESs Can Reflect the Effectiveness of Ecological Protection

After synthesizing the spatial and temporal distribution of the four ESs, the areas with high ecological levels in Gannan are mainly located in the southwestern counties of Maqu and Luqu, as well as in the southeastern southern counties of Diebu and Zhouqu, which are dominated by grasslands, wetlands, and forested lands, sparsely populated, and rich in natural resources. On the other hand, Hezuo City, Xiahe County, Zhuoni County, and Lintan County in northern Gannan have relatively low comprehensive levels of ESs, driven by natural factors on the one hand, and human activities on the other hand, with large-scale urban expansion and tourism development causing a decline in the level of ESs in these areas.

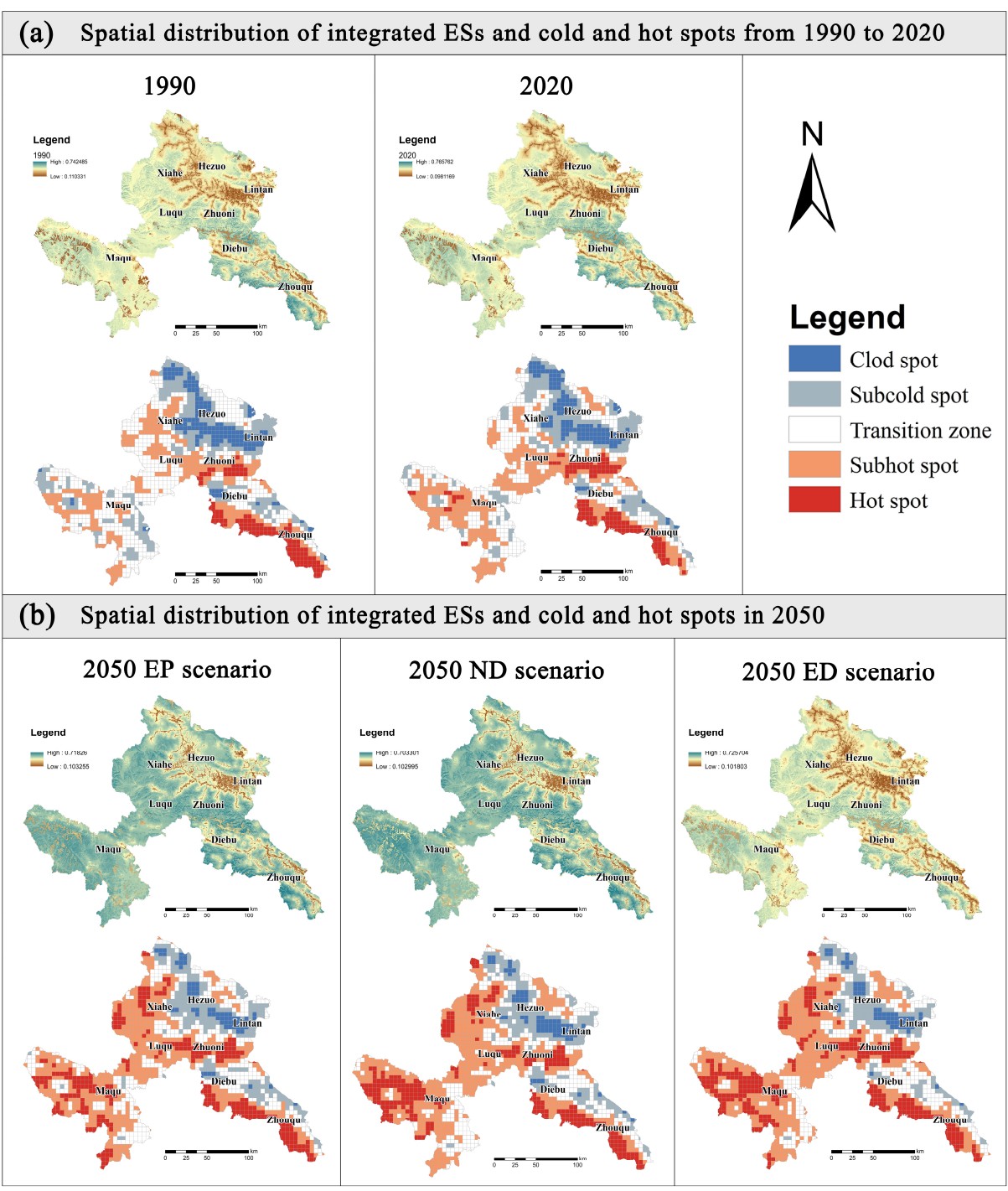

**Figure 9.** Spatial and temporal changes in the distribution of integrated ESs and hot and cold spots in Gannan from 1990 to 2050.

The success of Gannan's ecological management is shown in LUCC. The establishment of nature reserves can effectively control land use and prevent ecological damage caused by excessive human intervention [44]. In the Gahai Zecha National Nature Reserve, established in 1998, the quality of the habitat has been significantly improved, while sandy areas can be seen to have been virtually eliminated at the border between the Tibetan and Loess Plateaus (Figure 10). These achievements not only rely on the strong resilience of the ecosystem but also require the planning and management of government departments in conjunction with the active cooperation of the people.

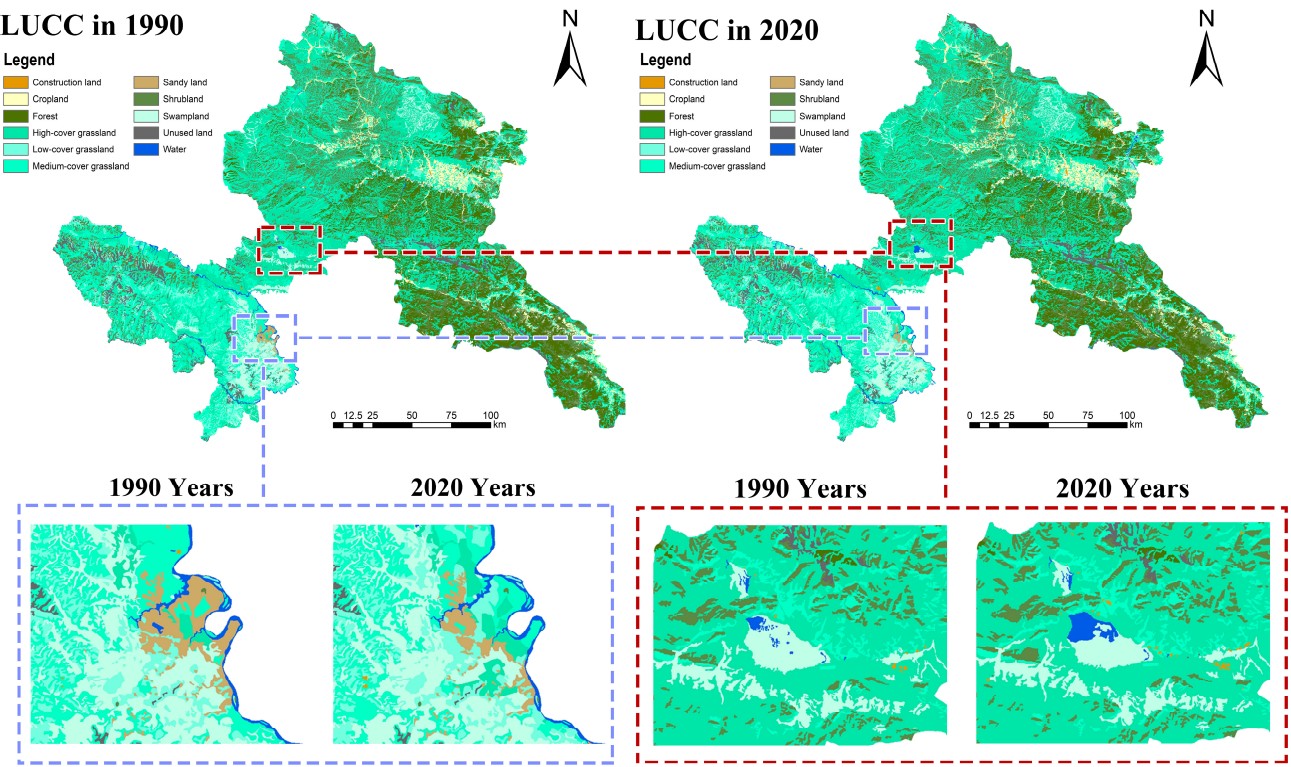

**Figure 10.** Detailed map of land use change in Gannan.

Nowadays, ESs and other related studies have been increasingly emphasized, and the scope of the study is becoming wider and wider, applicable to all kinds of areas at different scales and in different natural environments [45]. In the course of this study, exploring LUCC was a key factor in revealing changes in ESs. LUCC is largely influenced by human activities in today's environment of rapid socioeconomic development, and among many studies, modeling and assessing some key ESs based on land use change has become the approach that most researchers are willing to adopt.

For the healthy development of the ecological environment in protected areas, it is crucial to pay attention to LUCC and ESs, and it is necessary to conduct analysis and research from multiple perspectives and using various methods. For example, in a study of the Jiroft Plain in Iran, it was found that the relationships between multiple ESs undergo abrupt changes over time, and are directly related to LUCC. Therefore, when formulating and considering ecological strategies, it is not sufficient to focus solely on one ES indicator at a time; instead, multiple indicators need to be balanced to minimize ecological risks as much as possible [46]. When studying areas dominated by forests and grasslands, it is equally important to incorporate the experience of forest ecological classification and vegetation dynamic assessment. This can provide valuable insights into the impact of land use change on plant community diversity and ecosystem services [47]. Some studies have also incorporated economic and trade-related factors, spanning multiple countries, to explore the coupling relationships among land use, ecosystem services, and socioeconomic factors at larger scales [48].

Compared with this study, most other studies focus on regions with higher economic development or more frequent human activities. Constrained by diverse terrain and local socioeconomic factors, the study area of this paper experiences relatively low human activity, with forests and grasslands as the dominant land use types rather than croplands and urban areas; however, it still faces many challenges imposed by nature itself. The simulation and evaluation models we adopted are highly suitable for the ecological environment of Gannan. Furthermore, in conjunction with SSP-RCP data, the simulation of future scenarios is more objective. Nevertheless, it is still necessary to comprehensively assess the

spatiotemporal changes in ecosystem services and adjust our ecological strategic policies accordingly. Therefore, in future research, more detailed data precision, more accurate interpretation modes, and more advanced simulation models can make our research more scientific and practical. To a certain extent, combining LUCC and ESs and highlighting their coupling relationship might serve as a source of inspiration and guidance for the ecological environment's sustainable growth.

*4.3. Insights and Recommendations for Ecological Strategies and Management in Gannan*

It is very meaningful to analyze and discuss the practical significance behind the numerical values in the context of this study, and to put the theory and results into practice to address the requirements of ecological and economic sustainability in Gannan, so as to propose some decision-making support at the level of ecological protection [1]. Therefore, in order to strengthen the ecological environmental protection in Gannan, some recommendations on regional policies and management of the study area are given. These suggestions can also serve as a reference for ecological conservation strategies in economically underdeveloped areas.

(1) First, it is important to consider the function of the ecological security barrier in Gannan, as the ecological level of this area will impact more ecological environments within the watershed [32]. Protecting and managing water resources and enhancing the level of water conservation are the basic requirements for enhancing the sustainable development of the local ecology and economy, as well as the key links that affect the ecological security of the entire basin, especially the establishment of a conservation policy for grasslands and wetlands to minimize anthropogenic interventions and to safeguard the security of the Yellow River Basin's ecology [49].

(2) Second, Gannan is a region with pleasant scenery and a wealthy culture, especially where the prosperity of tourism has brought new opportunities and challenges to Gannan. In many developing regions, the growth of tourism is an inexorable trend that has enormous implications for both ecological and local economic development [50]. As a result, in order to integrate the sustainable growth of the natural environment with economic development, we need to manage the land use in line with planning requirements; to ensure that the nature reserves are not infringed upon; to strictly establish the red line of ecological security; to scientifically assess whether the project will pose a threat to the ecosystem, as economic development will often bring pressure on the natural environment; to weigh the synergistic pros and cons of the project; and to prioritize ecology.

(3) Finally, governments and administrators should emphasize the importance of ecological environmental protection for local residents and promote sustainable local development by combining scientific theories and expertise with domestic and international experience. Regarding Gannan, an administrative region characterized by a multitude of ethnic communities and distinctive customs and cultures, the preservation of the natural world encourages everyone to get involved, so the relevant policies for environmental protection need to be adapted to the local conditions, and the formulation of management policies should be carried out in depth in the life and production of the people in Gannan, fully examining the influences and dependence of human beings on nature and nature on human beings, and combined with the results of the research data, policy formulation for land planning and ecological protection should be carried out.

## 5. Conclusions

The land use data from historical scenarios were utilized to produce land use data for three future ecological protection scenarios by coupling the PLUS and InVEST models, and finally, through the correlation analysis and spatial and temporal change analysis of cold spots and hot spots, the main influencing factors driving LUCC and ESs were derived.

(1) Land use transfer in Gannan during the period 1990–2020 mainly occurred in the southwest and northern regions, with obvious spatial heterogeneity. In the historical context, the grassland land type occupies the largest area of 20,808.07 km², followed by forested land, with an area of 11,257.73 km², and sandy land. The smallest area was 70.59 km². Transfer between land use types mainly occurred between forested land, cropland, and grassland. With the current development model and planning objectives in Gannan, this trend of change may continue, posing challenges in balancing the demand for and security of food with the protection of forests and grasslands. These issues are also challenges faced by many countries and regions.

(2) Based on the new combination of RCP and SSP, three kinds of future scenarios were set up. Under the scenario of EP, the shrub forests are greatly increased, and the vegetation cover is shifted to a higher degree of coverage; under the scenario of ND, the degrees of likeliness of land use basically conform to the prediction of the Markov chain with relatively minor changes; and under the scenario of ED, the change in land use is mainly reflected in the large increase in the land used for construction. In reality, the transformation of land use is influenced by multiple uncontrollable factors. Currently, our research can only rely on different scenario simulations to predict future development trends. The results reveal significant differences in spatial distribution patterns, but there remains uncertainty in the actual transformation of land use in the future.

(3) Evaluation and analysis of ESs in Gannan through the InVEST model showed that the places with high values of water yield were mostly dispersed throughout Maqu County's western region and the junction of Diebu and Zhouqu Counties, and were relatively small in Gannan's central and northern regions; the places with high values of carbon storage were distributed in forests and shrub forests, with the overall pattern being very similar to that of the land use distribution pattern; the geographical distribution of soil retention indicated a high in the west and a low in the east; and the quality of the habitats was relatively lower in the towns and cities where there were higher population densities, and the indices of the quality of the habitats were higher in the regions where there was less intervention by human beings.

(4) The spatial and temporal transformations of ESs are influenced by multiple effects of natural and socioeconomic factors, and are correlated with most of the factors, and there are obvious tradeoffs and synergies. Between 1990 and 2020, the clustering of the distribution of integrated ESs became more and more significant; the cold spot areas are mainly concentrated in Gannan's northern region, and the hot spot areas are mainly concentrated in Gannan's southern and southeastern regions. Under the different scenarios in 2050, the highest integrated level of ESs is in the EP scenario, while the lowest level is observed in the ED scenario. These results can provide a certain reference for terrestrial ecosystems with similar climate types and geographical environments.

**Author Contributions:** Conceptualization, X.L.; methodology, X.L. and Y.L.; software, X.L., F.L. and Y.Z.; validation, Y.L.; formal analysis, H.Z. and J.Z.; investigation, X.L., F.L. and Y.Z.; data curation, H.Z. and J.Z.; writing—original draft preparation, X.L.; writing—review and editing, X.L.; visualization, X.L.; supervision, Y.L.; funding acquisition, Y.L. All authors have read and agreed to the published version of the manuscript.

**Funding:** This study was funded by the National Natural Science Foundation of China (32160409).

**Institutional Review Board Statement:** Not applicable.

**Informed Consent Statement:** Not applicable.

**Data Availability Statement:** The data presented in this study are available on request from the corresponding author.

**Conflicts of Interest:** The authors declare no conflict of interest.

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
