# Peer review of "Spatiotemporal Variation and Prediction Analysis of Land Use/Land Cover and Ecosystem Service Changes in Gannan, China"

_sustainability, doi:10.3390/su16041551_

Round 1

Reviewer 1 Report

Comments and Suggestions for Authors

The anthropogenic alteration of natural ecosystems and the expansion of disturbed areas is a pressing global problem. From this point of view, this paper is relevant, as it draws attention to an important problem.

 The research has a scientific novelty. The integrated valuing of ecosystem services and tradeoffs (InVEST) model and the patch-generating land use simulation (PLUS) model are combined in this work to assess the spatiotemporal variance of ecosystem services in Gannan.

 This study has theoretical and practical significance. This study contributes to the analysis of the developmental traits of Gannan ecosystems and can serve as a model for the preservation of terrestrial ecosystems with comparable environmental traits.

 There are comments to the annotation. The abstract does not reflect the results obtained well. This needs to be improved.

 In the introduction, the relevance of the study is well substantiated is described. The research objectives are formulated clearly and clearly. The description of the current state of the problem can be improved. The current version of the paper has regional specifics. It is necessary to reconsider the formulation of the task and the understanding of the results so that the paper becomes more interesting to readers from other countries, not just from China. It is necessary to put the necessary accents.

The methodology approaches are described in detail. The authors used methods adequate to the tasks set. Adequately selected methods allow us to consider the conclusions justified.

 The research results are illustrated with figures. The paper contains 9 visual figures and 3 informative tables. The results are presented clearly and clearly.

The current version of the discussion should be improved. The Discussion section is an important part of the paper. It is here that the authors should determine the place of their research in world science, compare the results obtained with those of other studies, including those carried out in other countries and climatic zones, explain the strengths of the approaches used, the novelty, the practical and theoretical significance of the results  in comparison with other studies. The authors practically do not include in the discussion the research developments that have been received so far. For example, even a single journal, Land, has many articles devoted to this problem for other regions. It is useful to compare research findings and practical recommendations. In addition, it is highly desirable to compare the research approach and results with those of other countries to identify similarities and differences in research methodology and results. For example, compare the environmental classifications that underlie the study and management of land use. You can rely on the review article on this topic: Experience of Forest Ecological Classification in Assessment of Vegetation Dynamics. Sustainability 2022, 14, 3384. https://doi.org/10.3390/su14063384
It is also possible to discuss the results obtained from the point of view of the problem of environmental indicators. It is also appropriate to compare the system of environmental indicators used in the paper with the systems of environmental indicators used by other authors, including those from other countries.
 Conclusions follow from the results and are reasonable.  However, the conclusions should also be improved so that they are more interesting to readers from other countries. The paper will be of interest to a wide range of readers whose scientific interests are related to land use, ecosystem services and sustainable regional development. Despite the fact that English is not my native language, I read the paper with interest and had no difficulties in understanding.

Reviewer 2 Report

Comments and Suggestions for Authors

I appreciate the article and its findings. There are some parts that explain the method and the stages of process that could be better explained, how they work or what is their purpose in the whole research. At some point I was not clear where the study was going.

Another issue is with the pictures that have a low resolution and the differences between them are not very well visible. The most relevant image from the whole article is figure 8 where you can see clearly the differences on the pixeled maps. Other images are not clear enough or big enough to be relevant. Maybe a map with differences would be more appropriate. Another improvement could be visual graphic to make a better understanding of the information from the tables (table 1) and Figure 7 should be explained better.

The recommendations and the conclusion are ok and open future possibilities for testing scenarios that try to offer solutions for the ecosystem.

Comments on the Quality of English Language

Some misspells or caption problems are present and should corrected. Some phrases are hard to follow and understand and should be rewritten (382-389 for example but there are others as well)

Reviewer 3 Report

Comments and Suggestions for Authors

The article provides a reasonable overview of the focus on land use/land cover changes (LUCC) and their implications for ecological risks and socio-economic development. It also highlights the significance of ecosystem services (ESs) and the role of simulation models in studying LUCC. It discusses the spatial and temporal variation of ecosystem services (ESs) in Gannan, with a focus on water yield, carbon storage, soil retention, and habitat quality. Finally, it applies historical land use data scenarios to model future ecological protection scenarios using two distinct models (PLUS and InVEST).

The manuscript is overall well written and recommended for publication after minors corrections to be made as follows:

1. Reduce the use of repetitive phrases and redundancies, such as repeated emphasising the importance of LUCC and ESs in the Introduction section. It is recommended to streamline the language and avoid unnecessary repetition.

2. Certain terms, such as "abstract ecosystem service functioning" and "cold hot spots of ESs," need clearer definitions or explanations. Please provide concise definitions or references to literature to enhance the precision of the terminology.

3. The description of of the PLUS model, and the LEAS and CARS components are too brief and require the reader to search for detailed information. Please improve this section as it will give a better understanding of the processes involved in this study.

4. Figure 6 should be quality improved. Particularly the proportion between the different font sizes. Letters that name the horizontal panels (a, b, c, d) look too large with respect to the internal wording (which in turn are critical for proper understanding of the information content).

5. Same for Figure 9. 

6. The manuscript lacks of comparative examples from similar studies over other regions of the globe. Is the methodology proposed in this study completely new? If yes, it has to be clearly stated in the document. Otherwise, the authors should include a review of similar studies and indicate in the conclusions how the current work compares with them.

Comments on the Quality of English Language

The overall quality of the English language is generally good. The text demonstrates a complete command of English and ideas are expressed clearly. However, there are instances of minor grammatical issues such as awkward phrasing, inconsistent verb tenses, and occasional use of complex sentence structures that can slightly make difficult the readability. Overall, the language clarity is enough, but a careful review of coherence and grammatical coherence would help.

Round 2

Reviewer 1 Report

Comments and Suggestions for Authors

The authors have responded to all my comments. I have no further comments.